# Extra-Gynecological Pelvic Pathology: A Challenge in the Differential Diagnosis of the Female Pelvis

**DOI:** 10.3390/diagnostics12071693

**Published:** 2022-07-12

**Authors:** Betlem Graupera, Maria Ángela Pascual, Stefano Guerriero, Jean Laurent Browne, Beatriz Valero, Silvia Ajossa, Serena Springer, Juan Luis Alcázar

**Affiliations:** 1Department of Obstetrics, Gynecology and Reproduction, Hospital Universitari Dexeus, 08028 Barcelona, Spain; betgra@dexeus.com (B.G.); marpas@dexeus.com (M.Á.P.); juabro@dexeus.com (J.L.B.); beaval@dexeus.com (B.V.); 2Centro Integrato di Procreazione Medicalmente Assistita (PMA) e Diagnostica Ostetrico-Ginecologica, Blocco Q, Azienda Ospedaliero Universitaria-Policlinico Duilio Casula, 09042 Monserrato, Cagliari, Italy; gineca.sajossa@tiscali.it; 3Department of Medical, Surgical and Health Sciences, University of Trieste, 34127 Trieste, Italy; serena.springer@gmail.com; 4Department of Obstetrics and Gynecology, Clínica Universidad de Navarra, School of Medicine, University of Navarra, 31008 Pamplona, Spain; jlalcazar@unav.es

**Keywords:** ultrasound, color Doppler, female pelvis, extra-gynecological disease

## Abstract

Ultrasound technology with or without color Doppler allows a real-time evaluation of the entire female pelvis including gynecologic and non-gynecological organs, as well as their pathology. As ultrasound is an accurate tool for gynecological diagnosis and is less invasive and less expensive than other techniques, it should be the first imaging modality used in the evaluation of the female pelvis. We present a miscellany of non-gynecological pelvic images observed during the realization of gynecological ultrasound. Transvaginal and transabdominal ultrasound is the first choice among diagnostic techniques for the study of the female pelvis, providing information about gynecological and extra-gynecological organs, allowing for an orientation toward the pathology of a specific organ or system as well as for additional tests to be performed that are necessary for definitive diagnosis.

## 1. Introduction

### 1.1. Ultrasound Technique

Ultrasound is a less invasive and less expensive diagnostic imaging technique than other diagnostic imaging modalities. Ultrasound is the imaging diagnostic technique of choice in the diagnosis of the female pelvis [1]. The American Institute of Ultrasound in Medicine proposed the use of ultrasound examinations before other imaging modalities when the evidence shows that ultrasound imaging is at least equally or more effective for the anatomic area evaluated [2]. For a long time, ultrasound has demonstrated its great value in establishing a gynecological diagnosis, compared with surgical findings [3]. Moreover, it has proven to be an accurate complementary imaging method in acute abdominal disorders, providing additional information as well as the final diagnosis in many cases [4]. The use of point-of-care ultrasound (POCUS) allows the clinician to perform the ultrasound scan both at the medical office or the patient’s bedside and after the physical examination, correlating images with the patient’s symptoms and evaluating any changes in real time [5]. Ultrasound can be performed using a transvaginal or transabdominal approach with or without color and/or power Doppler. Transvaginal ultrasound is performed with an empty bladder using higher frequency transducers, which provide a high-resolution image due to the proximity between the pelvic structures and the transvaginal probe. Transabdominal imaging requires a distended urinary bladder, which acts as an acoustic window, facilitating the evaluation of pelvic structures. Both techniques have some limitations: the anthropomorphic features of patients and a not corrected distended urinary bladder can limit transabdominal ultrasound. Transvaginal ultrasound is not feasible in cases of virginal patients and in other cases with the impossibility of vaginal examination [6]. Ultrasound imaging has the added advantage of real-time imaging, which allows pelvic organs to be scanned, correlating symptoms with specific pelvic anatomical locations, and provides a specific diagnosis of pelvic pathologies using different ultrasonographic modalities [1].

### 1.2. Female Pelvis Anatomy

The evaluation of the female pelvis includes the study of the uterus and both ovaries but also the evaluation of the bladder, rectum, and the vascular structures and ligaments of this area. Evaluation of the uterus, the central organ of the pelvis, should comprise the assessment of the uterine size and position, including the uterine body with the endometrium, the isthmus, and the cervix. The study of ovaries should include the visualization of both ovaries, generally in the ovarian fossa of what is related anteriorly by the medial umbilical ligament, the ureter and internal iliac posteriorly, and the external iliac vein superiorly. Both the uterus and the ovaries vary in size and characteristics throughout the menstrual cycle and according to menopausal status [6].

There are some papers that evaluate the use of ultrasound in the study of the female pelvis, with the majority related to gynecological diseases. In this paper, we present the value of pelvic ultrasound for diagnosing non-gynecological pathology, classifying these features in:(1)Imaging of the digestive system;(2)Imaging of the uro-renal system;(3)Vascular imaging;(4)Lymphatic imaging;(5)Neurological imaging;(6)Cutaneous images;(7)Others.

## 2. Imaging of the Digestive System

Evaluation of the most caudal segment of the digestive tract should be included in the study of the female pelvis because of its close relationship with the uterus, cervix, and vagina. In addition, during a pelvic ultrasound, it would be possible to demonstrate digestive diseases such as appendicitis, diverticulitis, mucocele, appendicular plastron, GIST, and colon cancer that should be known. In some cases, it could be difficult to make a differential diagnosis between some of these pathologies and endometriosis, which is a benign gynecological condition that can affect some pelvic organs such as the intestine, in addition to the genital system.

### 2.1. Appendicitis

Appendicitis is inflammation of the vermiform appendix, which can progress to an abscess, ileus, peritonitis, or death in case of the absence of treatment [7]. Patients with pelvic and or abdominal pain should first be evaluated using ultrasonography. After confirming the normality of gynecological organs, a careful evaluation of the pelvic area including the bowel should be performed. They are three clinical findings with the highest predictive value for diagnosing acute appendicitis: (1) right lower quadrant pain, (2) abdominal rigidity, and (3) migration of pain from the periumbilical region to the right lower quadrant [8]. Ultrasound imaging shows typically the thick-walled inflamed appendix and sometimes, appendicolith and a small periappendiceal fluid collection [9].

The presence of a non-compressible concentrically layered, blind-ending tubular structure in the right adnexal region with thickened and smooth appendiceal walls, marked hyperemia, and no sign of cog wheel or incomplete septum is consistent with acute appendicitis. Using these indicators, transvaginal ultrasound is useful in detecting an inflamed appendix and in distinguishing acute appendicitis from pelvic inflammatory disease (Figure 1, Figure 2 and Figure 3) [10]. Ultrasound should be considered as the first imaging tool in patients with abdominal or pelvic pain with suspicion of an appendicular inflammation due to the presence of specific signs, based on which a diagnosis of appendicitis may be conducted; thus, ultrasound is a wide availability technique, compared with other diagnostic imaging tools.

### 2.2. Appendicular Plastron

A plastron is a progressive form of acute appendicitis, with a low frequency (2–6%). A pelvic ultrasound allows the diagnosis by the presence of a mass in the right iliac fossa surrounding the small loops with the inflammatory signs. The evolution could be toward an abscess in cases of no treatment or ineffective treatment (Figure 4) [11].

### 2.3. Appendiceal Mucocele

Appendiceal mucocele is the accumulation of mucoid material in the lumen of the appendix. Although there are no differences between benign and malignant cases, it is crucial recognizing this entity in order to prevent the spillage of the mucus during surgery for avoiding grave complications such as pseudomyxoma peritonei [12].

The appendiceal mucocele shows at ultrasound a well-defined, thin-walled cystic mass with an ovoid, oblong, or pear-shaped morphology. The echogenicity of a mucocele depends on the composition of the mucus, ranging from an anechoic or hypoechoic cyst with low-level internal echoes and posterior acoustic enhancement to a heterogeneous avascular mass without posterior acoustic enhancement. Mucocele often shows a pattern of concentric echogenic layers within a cystic mass called the “onion skin” pattern, and it is fairly specific for appendiceal mucocele. Calcifications may be present in the wall (Figure 5, Figure 6 and Figure 7) [13].

### 2.4. Diverticulosis and Diverticulitis

Transvaginal and transabdominal ultrasound may be useful for the evaluation of diverticulitis, most cases of which occur in the sigmoid colon, which is in the pelvis.

In normal conditions, colonic diverticula are not evident at ultrasonography because of the colon’s thin wall. However, in cases of an inflamed diverticulum, it has a hyperechoic shadowing center surrounded by a thick hypoechoic edge, which represents the thickened wall. The wall is surrounded by inflamed hyperechoic paracolic fat. The characteristic hypertrophy of the circular muscle layer of the muscularis propria in cases of colonic diverticulosis is evident on ultrasound as circumferential thickening of the outer hypoechoic layer of the colon wall (Figure 8) [14].

### 2.5. Bowel Tumors

Tumors that affect the small intestine, cecum, sigmoid colon, and rectum may be seen on transvaginal sonography. Colorectal cancer is the third most common cancer in the Western Hemisphere, and the incidence increases with increasing age. Screening has been shown to reduce colorectal cancer incidence and mortality, but organized screening programs are still to be implemented in most countries [15]. In this context, it is necessary to keep in mind that we can find the presence of colorectal cancer incidentally during an ultrasonographic study of the pelvis.

The ultrasonographic image varies according to the type of tumor: carcinomas and lymphomas generally appear as a focal mural thickening, whereas mesenchymal tumors and metastases appear as intra- or extraluminal localized round solid masses that are hypoechoic or heterogeneous (Figure 9).

Recognizing these characteristics, it is necessary to make a differential diagnosis with benign gynecologic diseases such as deep endometriosis affecting the recto-sigmoid bowel.

Gastrointestinal stromal tumors (GISTs) are rare tumors that account for a small percentage of gastrointestinal neoplasms. Gastrointestinal stromal tumor is the most common gastrointestinal mesenchymal tumor. Some GISTs develop outside the gastrointestinal tract, such as in the omentum, mesentery, and retroperitoneum, and this type of tumor is called an extra-gastrointestinal stromal tumor (eGIST). GISTs developing outside the stomach are associated with a higher malignancy potential. Generally, GISTs are an incidental, asymptomatic finding, but if GISTs present symptomatically, they can produce nausea, vomiting, abdominal distension, abdominal pain, or peritonitis. Ultrasound manifests as solid, non-uniform, pelvic, or abdominal tumors of mixed echogenicity with or without cystic areas, with abundant vascularization, and without acoustic shadowing. The presence of a tumor with these characteristics without connection to the intestinal wall and that does not originate in the uterus or adnexa is highly suspicious of being an eGIST (Figure 10 and Figure 11) [16,17].

## 3. Imaging of the Uro-Renal System

The evaluation of the kidneys, the bladder, and the ureters are mandatory in benign gynecologic diseases such as endometriosis affecting the anterior and posterior compartments, especially the uterosacral ligaments. In these cases, it is possible to diagnose pyelocaliceal dilation for ureteral obstruction.

In other cases, the incidental detection of urological findings such as megaureter, benign and malignant bladder pathology, pelvic kidney, and ureterocele may occur.

### 3.1. Mega Ureter and Ureterocele

The megaureter could be congenital or acquired. A diameter of a ureter more than 7–10 mm should be considered a megaureter (Figure 12) [18].

Ureterocele represents cystic dilatation of the intravesical segment of the ureter, it may be associated with either a single or double ureter. The congenital defect is the obstruction of the meatus, and the ureterocele is a hyperplasic response to this obstruction. Ureteral duplication is present in about 75% of patients with ureterocele [19].

The ultrasonographic image of the ureterocele is an anechoic cyst within the posterior aspect of the urinary bladder [20]. During the ultrasonographic exam, a non-obstructed ureterocele may vary its size by the flow of urine toward the urinary bladder (Figure 13 and Figure 14).

The differential diagnosis should be performed with tubal pathology as hydrosalpinx or pyosalpinx, which show septa or thickened mucosal folds, whereas ureterocele may demonstrate peristalsis [14].

### 3.2. Benign and Malignant Bladder Pathology

There is a wide range of benign (cystitis glandularis, cystitis cystica, intestinal metaplasia, nephrogenic adenoma, and endometriosis) and malignant lesions (adenocarcinoma in situ or invasive urothelial carcinoma with glandular differentiation), which can develop in the bladder.

Adenocarcinoma of the bladder is an uncommon tumor, the diagnosis of which is important in order to rule out the possibility of secondary involvement of the bladder by an adenocarcinoma from a different site [21].

Ultrasound shows whit the filled urinary bladder in cases of carcinoma an irregular solid, usually polypoid intravesical lesion, which presents highly vascularized (Figure 15) [22].

### 3.3. Pelvic Kidney

The pelvic kidney occurs when the kidney does not ascend normally to reach the renal fossa. Most are asymptomatic. The most frequent accompanying anomaly is reflux vesicoureteral. In the absence of other abnormalities, the prognosis for patients with renal ectopy is good. Their incidence range between 1:12,000 and 1:900 cases [23].

In most cases, an ectopic kidney appears in the pelvis as a solid mass, and it could be confused with a tumor of gynecological origin, so it is extremely important to know its presence to make a differential diagnosis (Figure 16 and Figure 17).

## 4. Vascular Imaging

The knowledge of vascular anatomy structures is fundamental to understanding some imaging, especially the color Doppler study of the pelvis.

The abdominal aorta is divided into two common iliac arteries at the level of the fourth–fifth lumbar vertebra. These arteries are divided into external and internal iliac arteries at the level side of the sacroiliac joint. The external iliac artery supplies the lower limb, and the internal iliac artery mainly supplies the pelvis. The internal iliac artery passes medially over the pelvic brim and descends into the pelvic cavity. At the upper margin of the greater sciatic foramen, it divides anteriorly and posteriorly. The ureter is located on the medial aspect of the internal iliac artery, and the pararectal space is seen between the ureter and the internal iliac artery. It is the main blood supply to the pelvic organs, the gluteal muscles, and the perineum, with the anterior and posterior trunks [24].

The uterine artery and vein are branches of the internal iliac artery and are located parallel to the uterus, and at the level of the isthmus, the uterine artery enters the uterus, forming the arcuate and radial branches. These uterine vessels anastomose with the ovarian vasculature in the pampiniform plexus, a network of arterial and venous flow suspended between the fallopian tube, ovary, and uterus [6].

### 4.1. Pelvic Congestion

Pelvic congestion is a cause of chronic pelvic pain, but it has also been reported in imaging studies of asymptomatic women. It is usually associated with intra-pelvic varicose veins in women with unexplained pain in the hypogastrium or pelvis that last’s more than 6 months. The nature of varicose pelvic veins is unknown, most certainly multifactorial, and due to mechanical and hormonal issues. Doppler ultrasound is widely used with good results. Diagnostic criteria include ovarian vein diameter larger than 4 cm, dilated and tortuous arcuate uterine vessels communicating with varicose pelvic veins, slow venous flow, and retrograde venous reflux (Figure 18) [25]. Due to its presence in asymptomatic women, this entity must be considered in any ultrasound study of the female pelvis.

### 4.2. Thrombosis

A thrombotic lesion may occur if an injury to the vessel wall or endothelial dysfunction, a decrease in blood flow, and thrombophilia of the blood (Virchow’s triad) are present. Systemic alterations in the hemostatic mechanism typically produce local thrombotic lesions in discrete segments of the vascular tree. Areas of decreased flow, stagnant zones, turbulent flow, and cell damage, including hemolysis, can lead to platelet adhesion and aggregation, as well as fibrin formation (Figure 19) [26].

Thrombosis of the ovarian vein is a rare occurrence. Ovarian vein thrombosis is most frequently observed in the puerperium state when Virchow’s triad is present. It can be associated with other conditions such as post-abortion infection, pelvic inflammatory disease, recent pelvic surgery, and gynecological cancer. Moreover, there are other risk factors such as alterations in various coagulation factors related to this entity. The color Doppler ultrasonography allows physicians to reach a diagnosis by demonstrating an anechoic or hypoechoic mass without flow on Doppler examination (Figure 20). Computed tomography and magnetic resonance imaging are useful in atypical presentations to avoid delays in diagnosis [27]. Transvaginal and transabdominal color Doppler ultrasound can demonstrate the normal anatomy of pelvic vascular structures as well as their pathology, and therefore, it is the technique of choice in some gynecological cases such as in the puerperium period, because it may perform differential diagnosis with other gynecological and non-gynecological pathologies.

### 4.3. Aneurysm

Isolated iliac aneurysms are rare, with a prevalence of around 2% of patients with aortoiliac aneurysms. Is less frequent in females than in males, with a ratio of 1:6, and more frequent in young women who have undergone pregnancy and delivery.

Around 25% of all patients present with a rupture of an isolated iliac aneurysm, resulting in a contained pelvic hematoma, which may result in compressive symptoms or intraperitoneal rupture. Ruptures into adjacent pelvic viscera including rectum and bladder and perianal ecchymosis have been reported because of dissection through the retroperitoneal tissues. Bleeding may be catastrophic, and the septic complications of surgery and subsequent mortality are likely high. Although the exact incidence remains unclear, it seems likely that a significant proportion of patients with an isolated iliac aneurysm are asymptomatic and have their aneurysm discovered on routine clinical examination or imaging (Figure 21) [28].

## 5. Lymphatic Imaging

Transvaginal and transabdominal ultrasound may show a range of lymphatic pathology, in cases of gynecological or extra-gynecological diseases.

Pathologic lymph nodes generally appear as solid lesions related to the iliac vessels (Figure 22 and Figure 23). In some cases, it may be necessary to use other imaging techniques, such as magnetic resonance or computed tomography, to determine the origin of these lesions.

Lymphocele is a collection of lymphatic liquid inside a thick, fibrous, non-epithelial wall and with no vascularization, arising from the retroperitoneum to the pelvic or abdominal cavity. Ultrasound shows lymphocele as an oval, round, or hourglass-shaped, and uni- or multilocular tumor with a thick wall. The contained fluid may be anechoic, low-level, ground-glass, hemorrhagic, or mixed echogenicity. It may present with intraluminal septations or debris (Figure 24) [29].

Cystic lymphangioma is a benign disease due to an incorrect embryological connection of the lymphatics in case of primary lymphatic cysts fail to converge with the main lymphatic system. It may develop in a wide range of anatomic lesions, with less frequency in the abdominal location. Differential diagnoses from a pancreatic pseudocyst, ovarian cyst, renal cyst, or other diseases should be carried out.

Ultrasound shows an abdominal unilocular or multilocular cyst with thin echogenic walls and hypoechoic or echogenic content. A hemorrhagic pattern is demonstrated in cases of internal bleeding (Figure 25) [30].

## 6. Neurogenic Imaging

Ultrasound provides images of neurologic structures, in some cases, as perineural cysts and neurogenic tumors.

### 6.1. Tarlov Cyst

The perineural cysts or Tarlov cysts appear on dorsal nerve roots, commonly in the sacral spine. Most are benign and asymptomatic, but occasionally, they can be symptomatic, producing compression. Tarlov cysts are incidental findings on MRI, which is considered the gold standard imaging technique [31].

In ultrasonography, Tarlov cysts are characterized by the presence of cystic lesions in one of both adnexal regions usually well-defined and sometimes completely anechoic or with the presence of some echoes or a network of very fine walls inside. It can be difficult to differentiate from other adnexal imaging such as hydrosalpinx, ureteral pathology, or pseudoperitoneal cysts (Figure 26).

### 6.2. Neurogenic Tumors: Schwannoma and Neurofibroma

Schwannomas are rare tumors that arise from the Schwan cells of the peripheral nerve sheath. Most cases are sporadic, but familial clustering may be seen in association with von Recklinghausen’s disease. Clinical features are non-specific, depending on the location and size of the lesion, the most common abnormalities of which are abdominal pain and neurological deficit. Radiological studies are fundamental for diagnosing Schwannomas [32].

Neurofibroma is another benign tumor composed of neoplastic Schwann cells, but unlike schwannoma, it also contains additional non-neoplastic components, including fibroblasts, mast cells, and perineurial-like cells, as well as residual axons. Sporadic cutaneous neurofibroma is the most common type, in contrast to neurofibromatosis type 1, which is characterized by the presence of multiple neurofibromas, in which there may be involvement of multiple cutaneous sites, peripheral nerves, and spinal roots (Figure 27) [33].

Although the MRI is the gold standard for diagnosing these diseases, the lesions caused by these tumors could be observed as lesions in routine pelvic scans.

## 7. Soft-Tissues Images

The pilonidal sinus is an acquired or congenital cutaneous disease that typically occurs in the intergluteal region and is mainly observed in young adults. Some authors demonstrated that ultrasound is useful for evaluating the margins, extensions, sinus tract, branches, and openings of the pilonidal sinus [34].

Transvaginal ultrasound demonstrates the typical characteristics of hair content in the cyst, showing mixed echogenicity and hyperechoic areas with acoustic shadows (Figure 28).

## 8. Others

There is a miscellaneous group of images that can be viewed in the pelvis. Within this group, it is important to recognize the presence of forgotten foreign bodies due to the legal consequences of their findings. Gossypiboma is the term used to describe the forgotten cotton/gauze piece in the body after surgical procedures. These foreign bodies, which are forgotten in the abdomen, produce ileus, intra-abdominal masse effect, postoperative abdominal pain, nausea, and vomiting. It should be differentiated from malignancies with mass-like images [35].

Ultrasound may show a well-delineated hypoechoic mass containing an echogenic structure with pronounced posterior acoustic shadows (Figure 29) [14].

## 9. Conclusions

The female pelvis is a complex anatomical region, in which a multitude of both gynecological and extra-gynecological images can be diagnosed.

Transvaginal ultrasound, complemented by transabdominal ultrasound, is the first choice among diagnostic techniques for the study of the female pelvis, providing information on the gynecological organs as well as the extra-gynecological organs.

This allows for an orientation toward the pathology of a specific organ or system as well as for additional tests to be performed that are necessary for a definitive diagnosis.

## Figures and Tables

**Figure 1 diagnostics-12-01693-f001:**
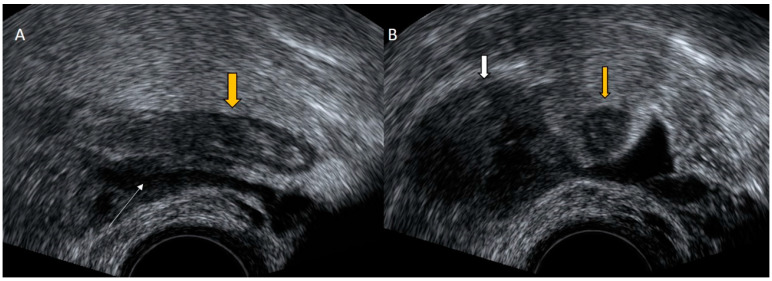
Grayscale transvaginal ultrasound of an inflamed appendix. Longitudinal transvaginal sonographic image demonstrates a blind-ending tubular structure in the right adnexal region pointed with yellow arrow and a small periappendiceal fluid collection (white arrow) (**A**). Transverse sonographic view showing the right ovary (white arrow) and the characteristic submucosal ring of an inflamed appendix (yellow arrow) (**B**).

**Figure 2 diagnostics-12-01693-f002:**
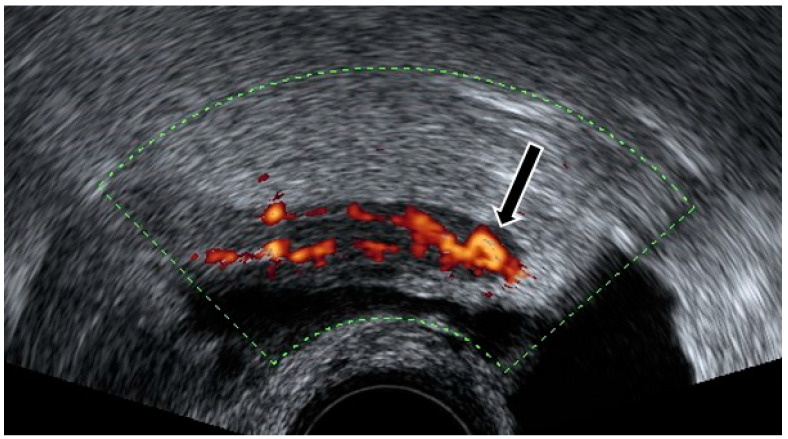
Power Doppler demonstrates hyperemia in the wall of the appendix (black arrow).

**Figure 3 diagnostics-12-01693-f003:**
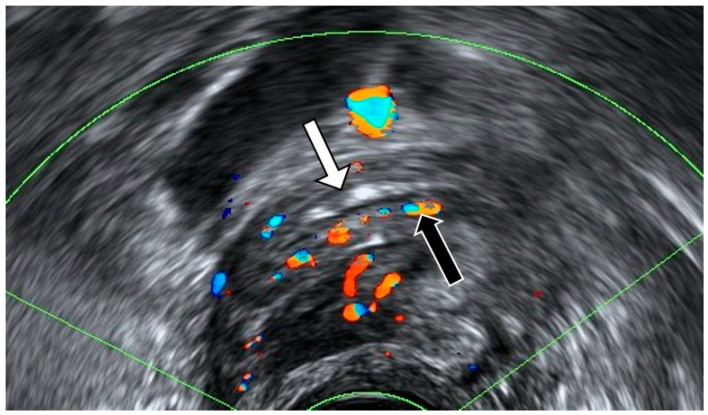
Longitudinal transvaginal ultrasound with color Doppler demonstrates the presence of intraluminal gas in the appendix (white arrow) and hyperemia (black arrow).

**Figure 4 diagnostics-12-01693-f004:**
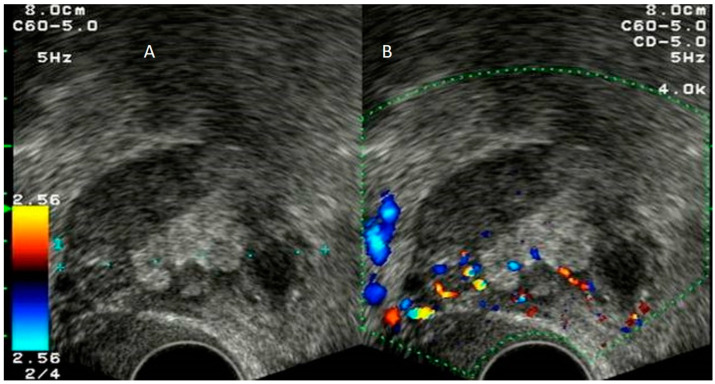
Grayscale (**A**) and color Doppler (**B**) of appendicular plastron showing a vascularized heterogeneous mass in the right iliac fossa.

**Figure 5 diagnostics-12-01693-f005:**
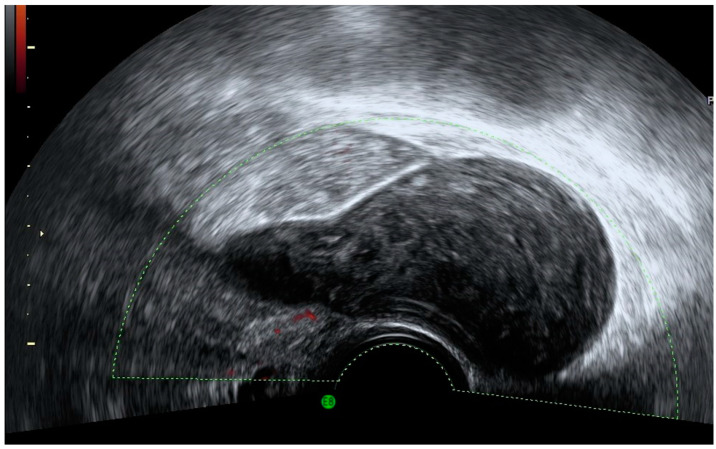
Power Doppler transvaginal ultrasound shows an appendiceal mucocele as a well-defined, thin-walled cystic mass with pear-shaped morphology with echogenic content and no vascularization.

**Figure 6 diagnostics-12-01693-f006:**
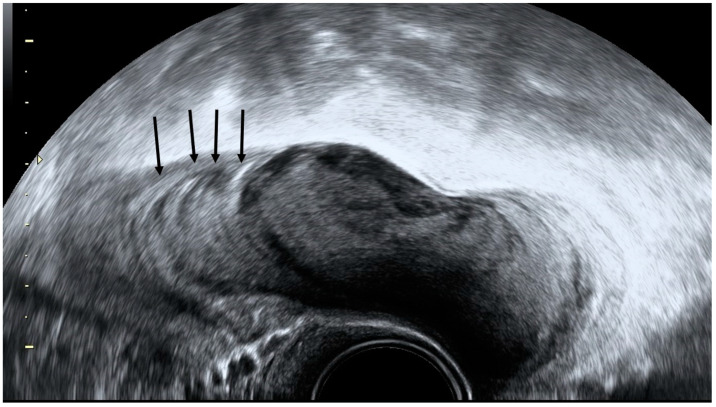
Sonographic features of an appendiceal mucocele. Ultrasound demonstrates the presence of characteristic concentric echogenic layers (arrows) within a cystic mass or “onion skin” pattern.

**Figure 7 diagnostics-12-01693-f007:**
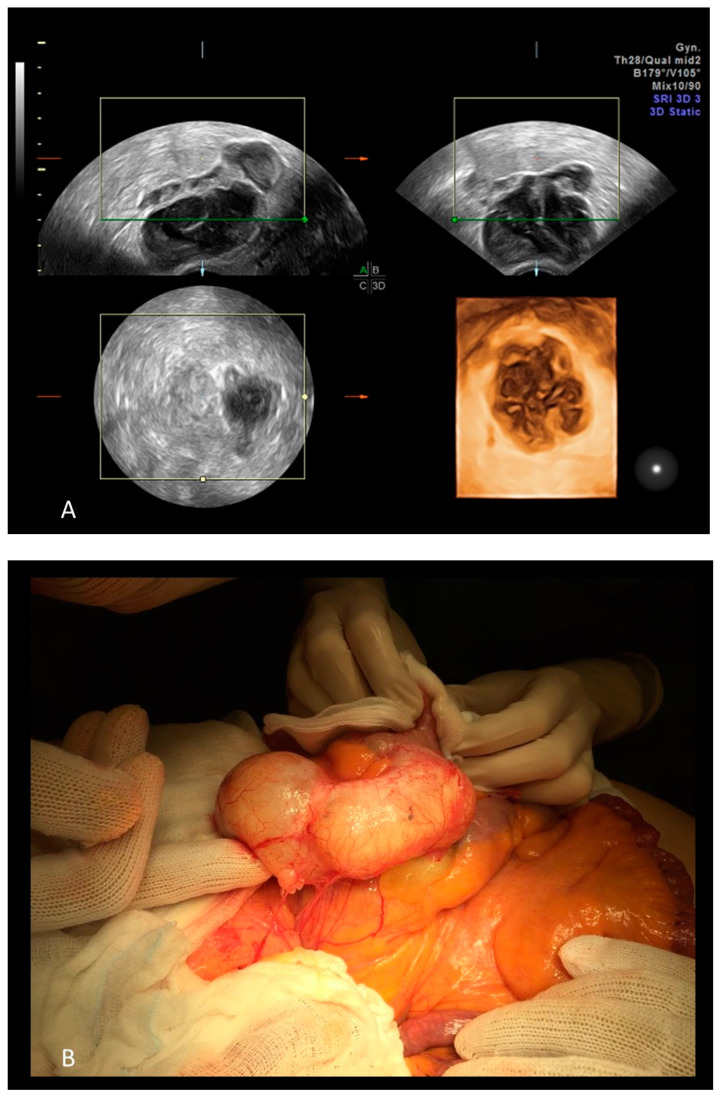
A three-dimensional rendering of a mucocele of 96 × 46 × 20 mm in an 80-year-old woman (**A**) and the same mucocele at surgery (**B**).

**Figure 8 diagnostics-12-01693-f008:**
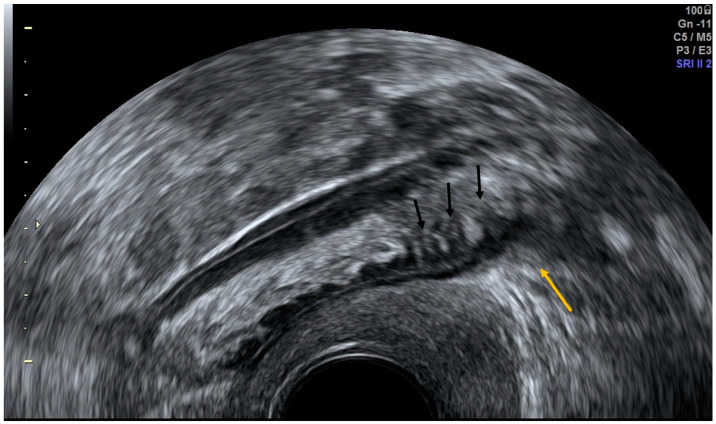
Transvaginal grayscale ultrasound in a patient with acute diverticulitis shows a hyperechoic image arising from the colonic wall with a hypoechoic rim representing wall thickening of the inflamed diverticulum (yellow arrow) and minimal wall thickening of the colon (black arrows).

**Figure 9 diagnostics-12-01693-f009:**
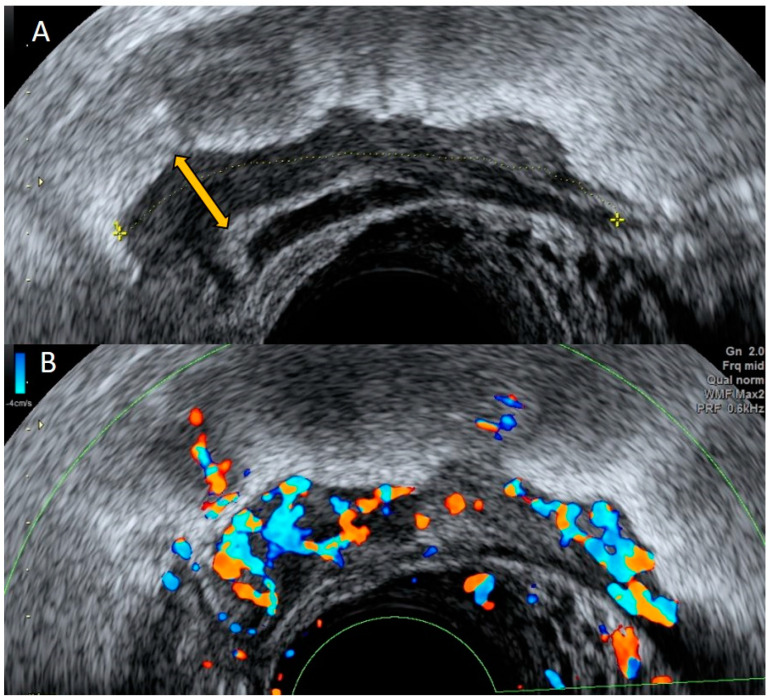
Transvaginal grayscale (**A**) and color Doppler image (**B**) of a patient with colonic carcinoma. Note the focal thickening of the colonic wall (double arrow) and the abundant vascularization in color Doppler study.

**Figure 10 diagnostics-12-01693-f010:**
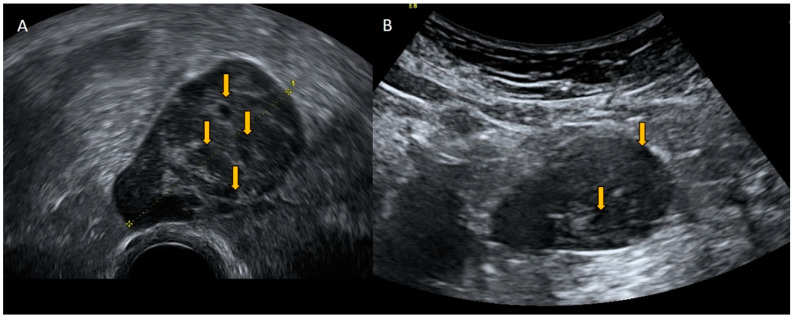
Transvaginal (**A**) and transabdominal (**B**) ultrasound of a patient with a GIST shows a solid pelvic tumor of mixed echogenicity with cystic areas (yellow arrows) and abundant vascularization and without acoustic shadowing.

**Figure 11 diagnostics-12-01693-f011:**
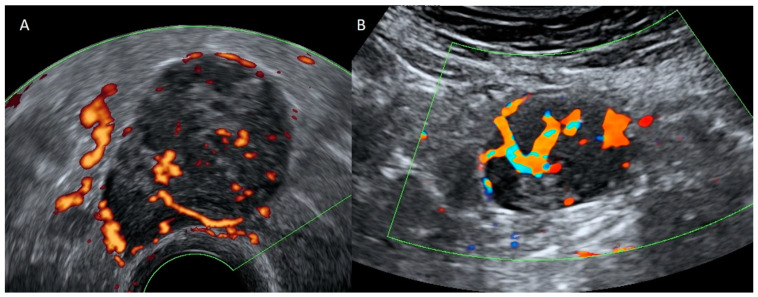
Power Doppler transvaginal ultrasound (**A**) and color Doppler transabdominal ultrasound (**B**) demonstrate abundant vascularization inside the GIST.

**Figure 12 diagnostics-12-01693-f012:**
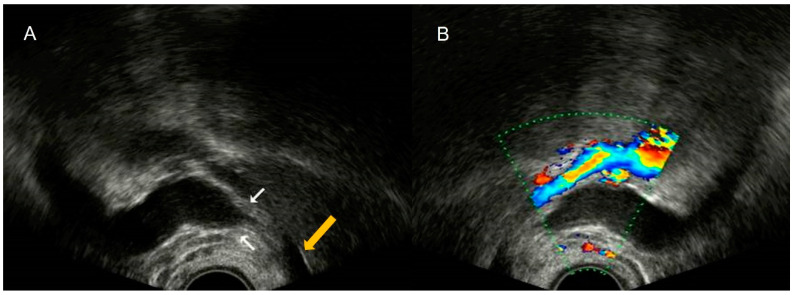
Ultrasound imaging demonstrates a megaureter with tapered distal segment (white arrows) ending at the bladder (yellow arrow) (**A**). Color Doppler shows no vascularization of megaureter (**B**).

**Figure 13 diagnostics-12-01693-f013:**
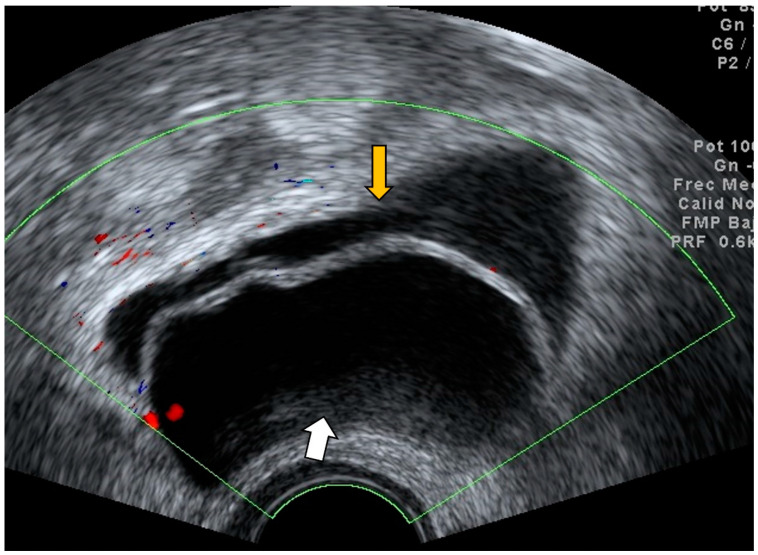
Ultrasonographic image of ureterocele showing an anechoic cyst (white arrow) within the posterior aspect of the urinary bladder (yellow arrow).

**Figure 14 diagnostics-12-01693-f014:**
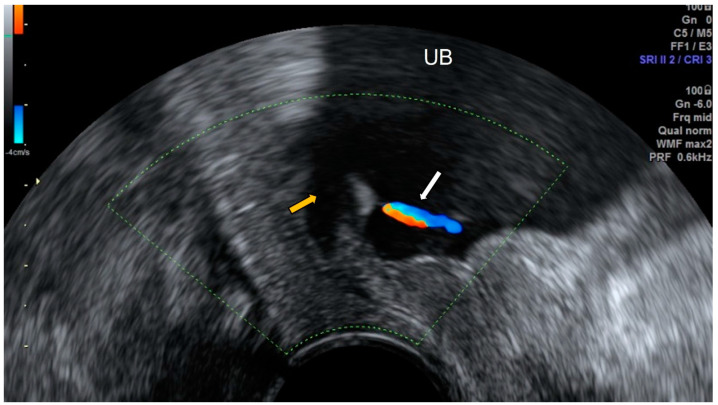
Transvaginal ultrasound shows a ureterocele (yellow arrow). Color Doppler demonstrates the flow of urine (white arrow) toward the urinary bladder (UB).

**Figure 15 diagnostics-12-01693-f015:**
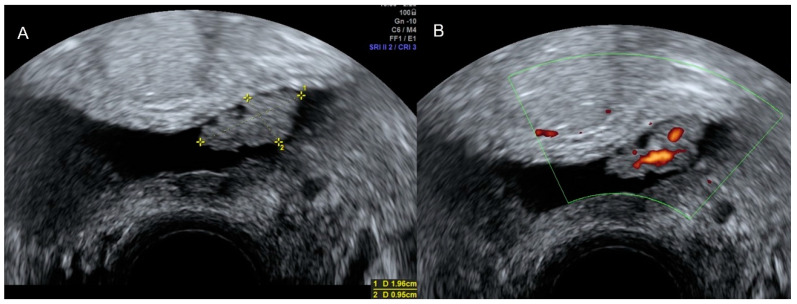
Transvaginal ultrasound shows the urinary bladder with an irregular solid, and polypoid intravesical lesion between the calipers (**A**). Power Doppler reveals high vascularization (**B**). Histology confirmed a bladder carcinoma.

**Figure 16 diagnostics-12-01693-f016:**
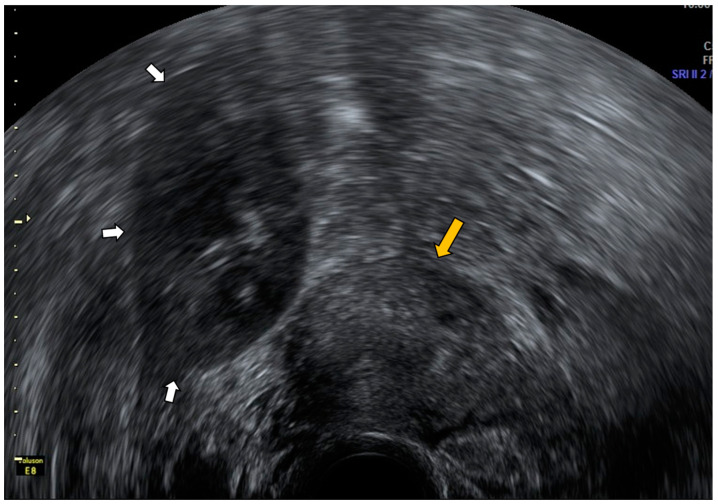
Transvaginal ultrasound shows a transversal view of the uterus (yellow arrow) and a solid mass in the right iliac fossa pointed with white arrows, presenting the ultrasonographic renal characteristics corresponding to an ectopic kidney.

**Figure 17 diagnostics-12-01693-f017:**
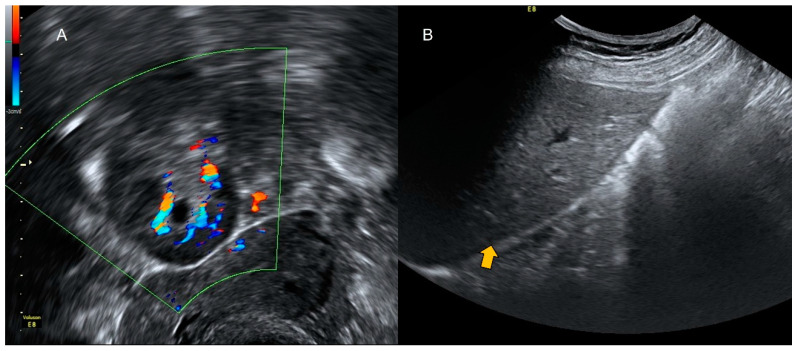
Color Doppler transvaginal ultrasound shows the vascularization of an ectopic kidney (**A**). Transabdominal ultrasound in the same patient demonstrates the presence of the liver (yellow arrow) and the absence of the orthotopic kidney (**B**).

**Figure 18 diagnostics-12-01693-f018:**
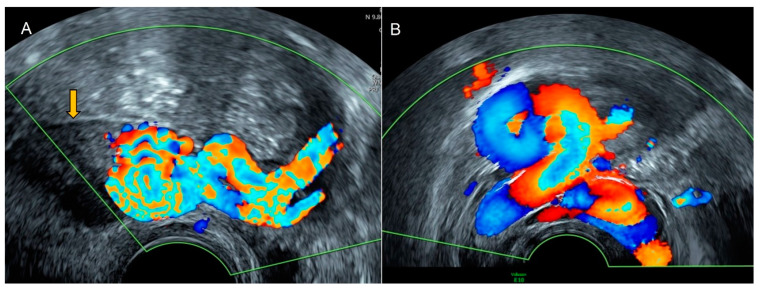
Color Doppler transvaginal ultrasound (**A**,**B**) demonstrates dilated and tortuous arcuate uterine vessels communicating with varicose pelvic veins. Uterus is indicated with yellow arrow (**A**).

**Figure 19 diagnostics-12-01693-f019:**
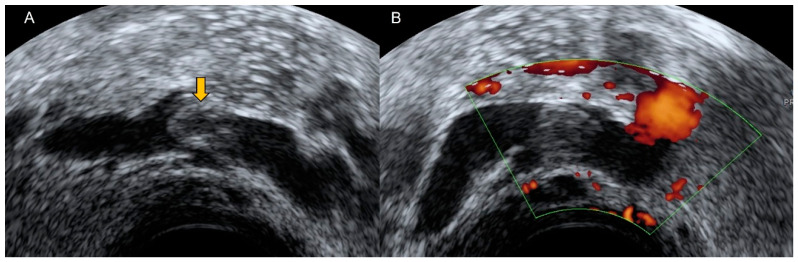
Grayscale (**A**) and power Doppler (**B**) transvaginal ultrasound show a uterine vein with the presence of a thrombus (yellow arrow) inside. Note the absence of vascularization in the image corresponding to the thrombus.

**Figure 20 diagnostics-12-01693-f020:**
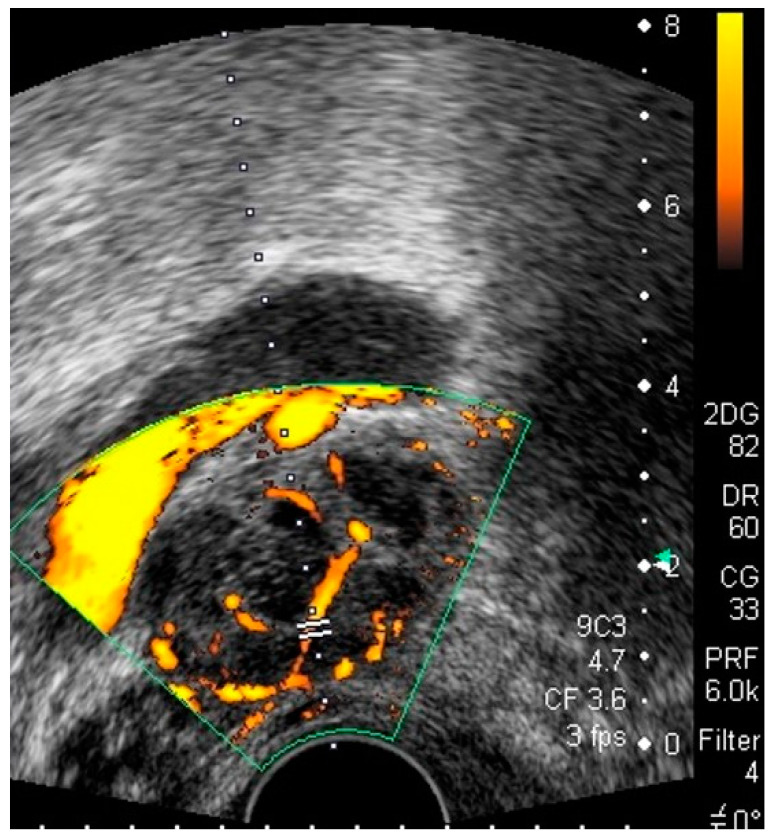
Transvaginal ultrasonography shows a septate cystic nodule independent from ovary corresponding to ovarian vein thrombosis. Power Doppler demonstrates peripheral and central vascularization.

**Figure 21 diagnostics-12-01693-f021:**
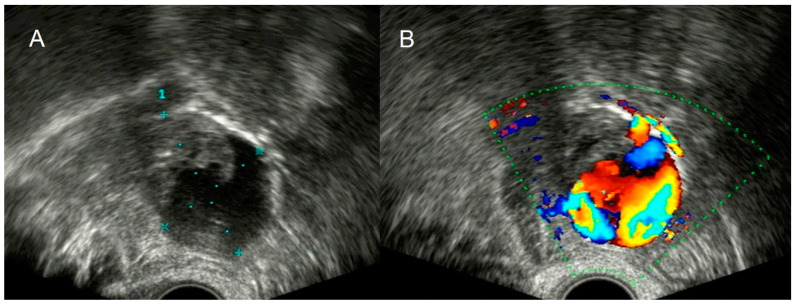
Grayscale transvaginal ultrasound shows a pelvic rounded cystic mass (**A**). Color Doppler evidence vascularization demonstrates the vascular nature of the lesion corresponding to a pelvic aneurysm (**B**).

**Figure 22 diagnostics-12-01693-f022:**
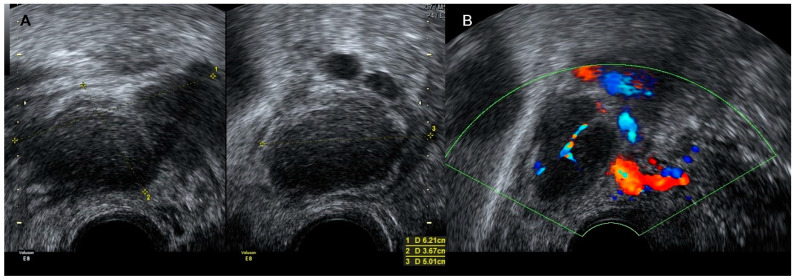
Transvaginal ultrasound shows an irregular nodule with diffuse echoes consisting of lymphoma (**A**). Color Doppler demonstrates central vascularization (**B**).

**Figure 23 diagnostics-12-01693-f023:**
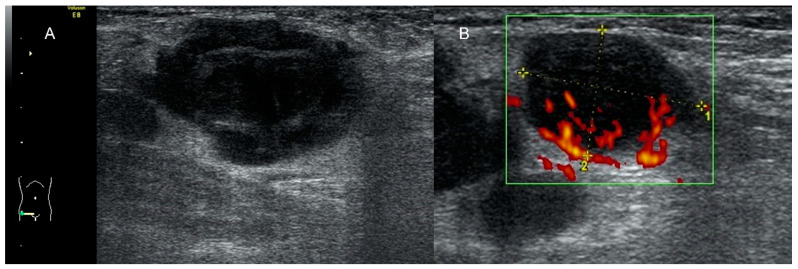
Soft-tissue ultrasound demonstrates an irregular hypoechoic nodule (**A**) in which power Doppler demonstrates abundant penetrating vascularization (**B**). Both Figure 22 and Figure 23 depict lymphoma.

**Figure 24 diagnostics-12-01693-f024:**
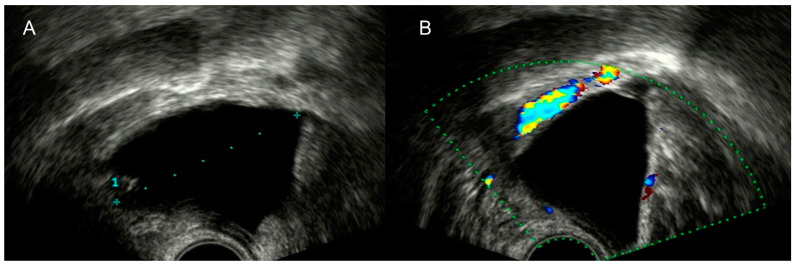
(**A**,**B**) Cystic lesion related to iliac vessels in a patient with previous lymphadenectomy consistent with lymphocele.

**Figure 25 diagnostics-12-01693-f025:**
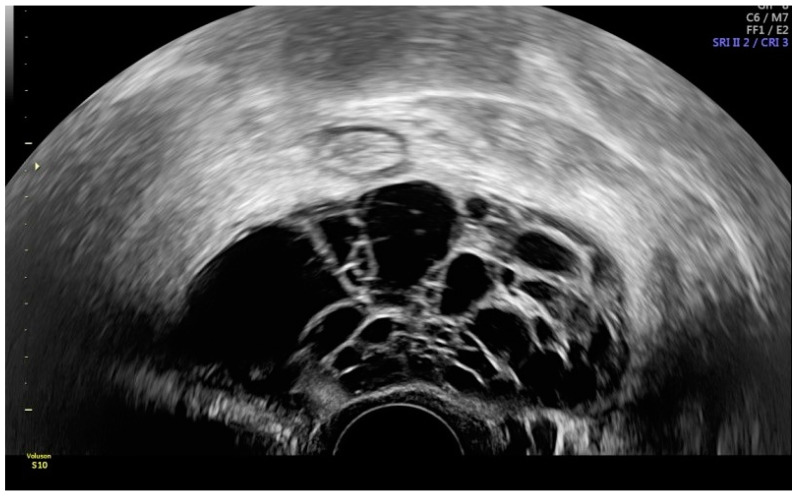
Transvaginal ultrasound shows a pelvic multilocular solid lesion independent from both ovaries in a patient with previous cystic lymphangioma.

**Figure 26 diagnostics-12-01693-f026:**
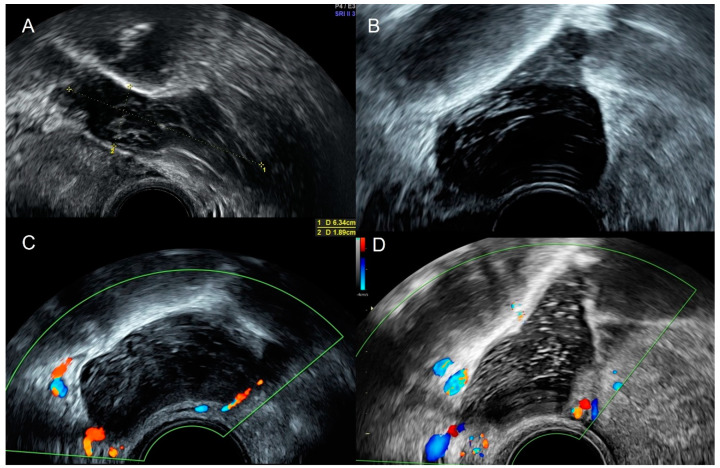
Transvaginal ultrasound shows Tarlov cyst as a well-defined cystic lesion in the adnexal region with the presence of some echoes or a network of very fine walls inside it (**A**–**D**).

**Figure 27 diagnostics-12-01693-f027:**
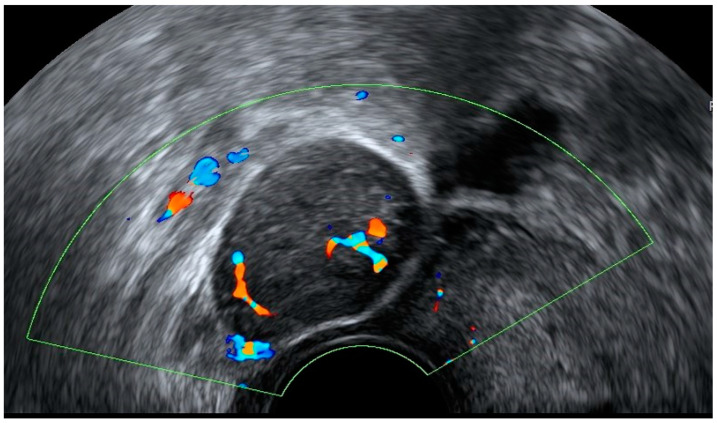
Color Doppler transvaginal ultrasound shows in the right adnexal region, a well-defined solid mass with cystic areas and scattered vessels in a patient with histological result of neurofibroma.

**Figure 28 diagnostics-12-01693-f028:**
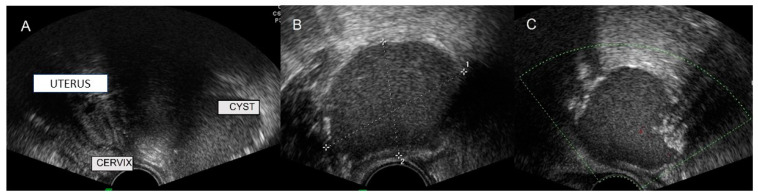
Transvaginal ultrasound sagittal view of the pelvis shows the uterus and the cervix and adjacent to the posterior wall, a cystic lesion with diffuse echoes, and hyperechoic images with acoustic shadows consistent with a pilonidal cyst (**A**–**C**). Power Doppler shows absence of vascularization (**C**).

**Figure 29 diagnostics-12-01693-f029:**
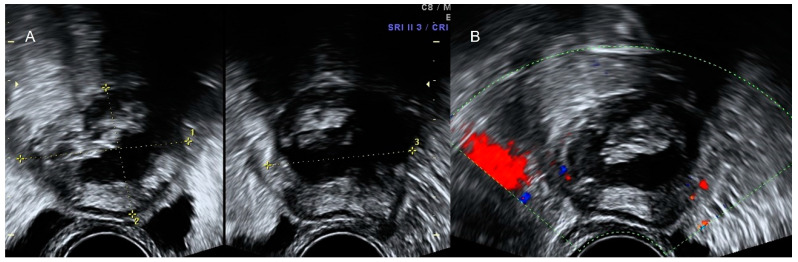
Transvaginal ultrasound imaging (**A**,**B**) shows a heterogeneous solid mass with acoustic shadows in a patient with recent pelvic surgery. Color Doppler does not demonstrate vascularization (**B**). This image corresponds to a surgical gossypiboma.

## Data Availability

The data presented in this study are available on request from the corresponding author. The data are not publicly available due to privacy law.

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
