# Peer review of "Extra-Gynecological Pelvic Pathology: A Challenge in the Differential Diagnosis of the Female Pelvis"

_diagnostics, 2022, doi:10.3390/diagnostics12071693_

Round 1
Reviewer 1 Report
The authors did the requested changes and thus, the value of their manuscript increased.
The strong point of the manuscript is represented by the beautiful and suggestive ultrasound images.
The term "UTERU" from figure 28A is still not corrected.
I recommend publication after this minor correction.
Author Response
The figure has been modified as correctly suggested.
Reviewer 2 Report
The specific revisions are appreciated.
Author Response
Thanks. All changes performed.
This manuscript is a resubmission of an earlier submission. The following is a list of the peer review reports and author responses from that submission.
Round 1
Reviewer 1 Report
Very valuable artice , worth to be published
Author Response
Reviewer 1:
Very valuable article, worth to be published.
Thank you very much for your comments. It is an honor for us these words.
Reviewer 2 Report
Specific descriptions of the sonographic images (figures) are excellent.
The variety of non-gynecologic entities which can be found in the pelvis represents valuable clinical information not previously described in detail, to my knowledge.
The inclusion of statements which describe sonographic findings which may be asymptomatic (e.g. pelvic congestion syndrome) but consistent with found pathology, is very important to recognize, adding value to the manuscript.
The authors may wish to add further clinical information, describing the potential diagnostic value of the sonography, in comparison with other imaging tools that may be more commonly used (e.g. in appendicitis). The Discussion can also include the particular clinical advantages that sonography has for documenting vascular pathology in the pelvis (e.g. ovarian vein thrombosis), underscoring its clinical value.
Author Response
Reviewer 2:
Specific descriptions of the sonographic images (figures) are excellent.
Thank you very much for this comment.
The variety of non-gynecologic entities which can be found in the pelvis represents valuable clinical information not previously described in detail, to my knowledge.
We try to include the vast majority of non-gynecological entities, which may appear in the context of a gynecological ultrasonography scan in our experience.
The inclusion of statements which describe sonographic findings which may be asymptomatic (e.g. pelvic congestion syndrome) but consistent with found pathology, is very important to recognize, adding value to the manuscript.
The performance of an imaging test such as pelvic ultrasound can show images that correspond to clinical entities that may or may not present symptoms but that are important to recognize. In this context, we wanted to also include these images that can sometimes present without clinical manifestations.
The authors may wish to add further clinical information, describing the potential diagnostic value of the sonography, in comparison with other imaging tools that may be more commonly used (e.g. in appendicitis). The Discussion can also include the particular clinical advantages that sonography has for documenting vascular pathology in the pelvis (e.g. ovarian vein thrombosis), underscoring its clinical value.
Thank you for this comment, which help to improve the manuscript.
We added this text in line 88:
“Due the presence of specific signs which may conduct to a diagnostic of appendicitis and being the ultrasound, a wide availability technique comparing with other diagnostic imaging tools, it should be considered ultrasound as the first imaging tool in patients with abdominal or pelvic pain with suspicion of an appendicular inflammation.”
We added this text in line 289:
“Transvaginal and transabdominal color Doppler ultrasound can demonstrate the normal anatomy of pelvic vascular structures as well as their pathology, being the technique of choice in some gynecological cases as in the puerperium period, because it may perform differential diagnosis with other gynecological and non-gynecological pathologies.”
Reviewer 3 Report
In the manuscript, the authors discuss differential diagnosis using images obtained by transvaginal ultrasound including Power Doppler. However, transvaginal ultrasound images provide medical staff with information about the presence, size, and location of a mass in the pelvis. Although the progress of medical technology is remarkable, it is very difficult to make a differential diagnosis or a definitive diagnosis with images obtained by transvaginal ultrasound. Ultimately, the differential or definitive diagnosis is conducted by surgical pathological diagnosis using surgically resected tissue or biopsy materials. However, the authors do not provide images obtained by surgical histopathology and/or molecular pathology in this manuscript.
In the title of the manuscript, the text "Extra-gynecological pelvic pathology" is mentioned, The authors do not provide images obtained by surgical histopathology and/or molecular pathology in this manuscript.
In this manuscript, the sentence "No patients have been directly involved, so patient informed consent is no needed" is described, but in this manuscript, so many images of patients obtained by transvaginal ultrasound including Power Doppler examinations are presented. For whatever reason In academic papers and conference presentations, when the medical staffs present clinical data, test data and images obtained from the patient, the medical staffs must provide all informed consent and all secondary informed consent obtained from all patients to participate in clinical research, etc.
In addition, the authors must submit a certificate of approval from the Institutional Review Board to the journal editorial.
Author Response
Reviewer 3:
In the manuscript, the authors discuss differential diagnosis using images obtained by transvaginal ultrasound including Power Doppler. However, transvaginal ultrasound images provide medical staff with information about the presence, size, and location of a mass in the pelvis. Although the progress of medical technology is remarkable, it is very difficult to make a differential diagnosis or a definitive diagnosis with images obtained by transvaginal ultrasound. Ultimately, the differential or definitive diagnosis is conducted by surgical pathological diagnosis using surgically resected tissue or biopsy materials. However, the authors do not provide images obtained by surgical histopathology and/or molecular pathology in this manuscript.
Thank you for your comments.
The purpose of this work is providing a miscellaneous of imaging demonstrating the value of transvaginal and transabdominal ultrasound for diagnosing pelvic non-gynecological diseases. In all cases of ultrasonographic malignant suspicion, surgery and histology confirmed the ultrasonographic diagnostic. In patients with ultrasonographic benign diagnostic, ultrasonographic follow-up, evaluation with other imaging techniques as MRI or TC and surgery in some cases confirmed the ultrasonographic suspicion. We recognize that the definitive diagnosis is surgical histopathology and/or molecular pathology but we did not provide these images because the objective of this work is showing specifically ultrasonographic imaging in non-gynecological pathology.
In the title of the manuscript, the text "Extra-gynecological pelvic pathology" is mentioned. The authors do not provide images obtained by surgical histopathology and/or molecular pathology in this manuscript.
With this title “Extra-gynecological pelvic pathology: a challenge in the differential diagnosis of the female pelvis” we demonstrate the value of ultrasound in the diagnosis of non-gynecological pathology. We recognize, as we explain in the previous comment, the surgical histopathology and/or molecular pathology as the definitive diagnosis but we want only to demonstrate the value of ultrasound compared with other imaging techniques as the first line diagnosis tool.
In this manuscript, the sentence "No patients have been directly involved, so patient informed consent is no needed" is described, but in this manuscript, so many images of patients obtained by transvaginal ultrasound including Power Doppler examinations are presented. For whatever reason in academic papers and conference presentations, when the medical staffs present clinical data, test data and images obtained from the patient, the medical staffs must provide all informed consent and all secondary informed consent obtained from all patients to participate in clinical research, etc.
All images of this work were obtained retrospectively from a database of images from patients, which signed an informed consent for using their data except their identification but including imaging data’s, thus we consider that a specific informed consent for this paper is no needed.
In addition, the authors must submit a certificate of approval from the Institutional Review Board to the journal editorial.
As we state, we consider that due to study’s design and nature, using retrospectively ultrasonographic images, IRB approval was waived. We include this sentence in the main text as the instructions for authors of the journal suggest.
Reviewer 4 Report
The manuscript represents a narrative review about the value of ultrasound examination in the differential diagnosis of pelvic conditions in women.
The real value of the manuscript is represented by the numerous and beautiful figures.
The abstract should include one key sentence from the conclusions.
The subchapter 1.1. Ultrasound technique from Introduction chapter should be developped and more references added.
The term POCUS should be mentioned, and some supplementary references added.
It is recommended that the authors mention the material and methods, even if the manuscript does not represent an original study: period of time, number of examinations and patients, number of examinators, technical specifications of the devices used, ethical disclaimer.
In figure 27 A, correct "UTERU".
Author Response
Reviewer 4:
The manuscript represents a narrative review about the value of ultrasound examination in the differential diagnosis of pelvic conditions in women.
The real value of the manuscript is represented by the numerous and beautiful figures.
Thank you for these comments.
The abstract should include one key sentence from the conclusions.
Thank you for your suggestion. We include it in line 22:
“Transvaginal and transabdominal ultrasound is the diagnostic technique of first choice in the study of the female pelvis, providing information about gynecological and extra-gynecological organs, allowing orientation towards the pathology of a specific organ or system and to perform the additional necessary tests for the definitive diagnosis.”
The subchapter 1.1. Ultrasound technique from Introduction chapter should be developed and more references added.
Thank you for the suggestion. We have modified this paragraph in line 30:
“Ultrasound is a less invasive and less expensive diagnostic imaging technique compared with other diagnostic imaging modalities.
And line 35: Ultrasound has demonstrated from a long time its great value in establishing a gynecological diagnosis, comparing with surgical findings [3](Loutradis D, Antsaklis A, Creatsas G, Hatzakis A, Kanakas N, Gougoulakis A, Michalas S, Aravantinos D. The validity of gynecological ultrasonography. Gynecol Obstet Invest. 1990;29(1):47-50. doi: 10.1159/000293299). Moreover, it has proven to be an accurate complementary imaging method in acute abdominal disorders providing not only additional information but also the final diagnosis in many cases [4] (Beyer D, Schulte B, Kaiser C. Ultrasound diagnosis of the acute abdomen. Bildgebung. 1993 Dec;60(4):241-7.).”
The term POCUS should be mentioned, and some supplementary references added.
We added a brief explanation about POCUS as well as a supplementary reference in the subchapter 1.1, line 39:
“The use of Point-of-care-ultrasound (POCUS) allows the clinician performing the ultrasound scan both at the medical office or the patient’s bedside and after the physical examination, correlating images with patient's symptoms, and evaluating any changes in real-time [5]. (Recker F, Weber E, Strizek B, Gembruch U, Westerway SC, Dietrich CF. Point-of-care ultrasound in obstetrics and gynecology.Arch Gynecol Obstet. 2021 Apr;303(4):871-876. doi: 10.1007/s00404-021-05972-5. Epub 2021 Feb 8.)”
It is recommended that the authors mention the material and methods, even if the manuscript does not represent an original study: period of time, number of examinations and patients, number of examinators, technical specifications of the devices used, ethical disclaimer.
Thank you for this recommendation, but we don't have finally included it in this pictorial ultrasound essay, considering that recently published pictorial essays in this journal (Guerriero S et al. Diagnostics (Basel). 2020 May 27;10(6):345. doi: 10.3390/diagnostics10060345; Cocco G et al. Diagnostics (Basel). 2021 Mar 29;11(4):609. doi: 10.3390/diagnostics11040609; Granata A. Diagnostics (Basel). 2021 Jan 11;11(1):101. doi: 10.3390/diagnostics11010101; Tamburrini S et al. Diagnostics (Basel). 2021 Feb 17;11(2):331. doi: 10.3390/diagnostics11020331; etc) do not include this section and because this is not a requirement of the instructions for authors of the journal for this type of articles.
In figure 27 A, correct "UTERU".
Done. Thank you.
Round 2
Reviewer 2 Report
Please look at Line 89 (Due to . . . ), and Line 91 (reword to "ultrasound should be considered as").
Altering citations was appropriate.
Author Response
Reviewer 2:
Please look at Line 89 (Due to . . . ), and Line 91 (reword to "ultrasound should be considered as").
Thank you.
We deleted (Due the presence of specific signs which may conduct to a diagnostic of appendicitis and being the ultrasound, a wide availability technique comparing with other diagnostic imaging tools, it should be considered ultrasound as the first imaging tool in patients with abdominal or pelvic pain with suspicion of an appendicular inflammation).
We reworded: Ultrasound should be considered as the first imaging tool in patients with abdominal or pelvic pain with suspicion of an appendicular inflammation due the presence of specific signs, which may conduct, to a diagnostic of appendicitis and being the ultrasound, a wide availability technique comparing with other diagnostic imaging tools.
Reviewer 3 Report
The authors have not modified the manuscript at all according to the comments from the reviewers.
In the manuscript, the authors discuss differential diagnosis using images obtained by transvaginal ultrasound including Power Doppler. However, transvaginal ultrasound images provide medical staff with information about the presence, size, and location of a mass in the pelvis. Although the progress of medical technology is remarkable, it is very difficult to make a differential diagnosis or a definitive diagnosis with images obtained by transvaginal ultrasound. Ultimately, the differential or definitive diagnosis is conducted by surgical pathological diagnosis using surgically resected tissue or biopsy materials. However, the authors do not provide images obtained by surgical histopathology and/or molecular pathology in this manuscript.
In the title of the manuscript, the text "Extra-gynecological pelvic pathology" is mentioned, The authors do not provide images obtained by surgical histopathology and/or molecular pathology in this manuscript.
In this manuscript, the sentence "No patients have been directly involved, so patient informed consent is no needed" is described, but in this manuscript, so many images of patients obtained by transvaginal ultrasound including Power Doppler examinations are presented. For whatever reason In academic papers and conference presentations, when the medical staffs present clinical data, test data and images obtained from the patient, the medical staffs must provide all informed consent and all secondary informed consent obtained from all patients to participate in clinical research, etc.
In addition, the authors must submit a certificate of approval from the Institutional Review Board to the journal editorial.
Author Response
Reviewer 3:
The authors have not modified the manuscript at all according to the comments from the reviewers.
We apologize to the reviewer for not including some of the suggested changes and would like to thank you for your review. We already indicated in our response to reviewers in the first round of reviews, the point-by-point responses to all comments.
In the manuscript, the authors discuss differential diagnosis using images obtained by transvaginal ultrasound including Power Doppler. However, transvaginal ultrasound images provide medical staff with information about the presence, size, and location of a mass in the pelvis. Although the progress of medical technology is remarkable, it is very difficult to make a differential diagnosis or a definitive diagnosis with images obtained by transvaginal ultrasound. Ultimately, the differential or definitive diagnosis is conducted by surgical pathological diagnosis using surgically resected tissue or biopsy materials. However, the authors do not provide images obtained by surgical histopathology and/or molecular pathology in this manuscript.
The purpose of this work is providing a miscellaneous of imaging demonstrating the value of transvaginal and transabdominal ultrasound for diagnosing pelvic non-gynecological diseases. In all cases of ultrasonographic malignant suspicion, surgery and histology confirmed the ultrasonographic diagnostic. In patients with ultrasonographic benign diagnostic, ultrasonographic follow-up, evaluation with other imaging techniques as MRI or TC and surgery in some cases confirmed the ultrasonographic suspicion. We recognize that the definitive diagnosis is surgical histopathology and/or molecular pathology (all the cases included have obviously a surgical and pathological confirmation) but we think that add more of these images is outside the topic of the paper. To avoid to further increase the length of the paper and the number of images (28) we just added two images (one with surgical findings).
In the title of the manuscript, the text "Extra-gynecological pelvic pathology" is mentioned, The authors do not provide images obtained by surgical histopathology and/or molecular pathology in this manuscript.
With this title “Extra-gynecological pelvic pathology: a challenge in the differential diagnosis of the female pelvis” we demonstrate the value of ultrasound in the diagnosis of non-gynecological pathology. We recognize, as we explain in the previous comment, the surgical histopathology and/or molecular pathology as the definitive diagnosis but we want only to demonstrate the value of ultrasound compared with other imaging techniques as the first line diagnosis tool. Anyway, we have added as at least one surgical image See new figures) also to avoid upsetting the manuscript primarily dedicated to ultrasonographic imaging.
In this manuscript, the sentence "No patients have been directly involved, so patient informed consent is no needed" is described, but in this manuscript, so many images of patients obtained by transvaginal ultrasound including Power Doppler examinations are presented. For whatever reason In academic papers and conference presentations, when the medical staffs present clinical data, test data and images obtained from the patient, the medical staffs must provide all informed consent and all secondary informed consent obtained from all patients to participate in clinical research, etc.
All images of this work were obtained retrospectively from a database of images from patients, which signed an informed consent for using their data except their identification but including imaging data’s, thus we consider that a specific informed consent for this paper is no needed.
In addition, the authors must submit a certificate of approval from the Institutional Review Board to the journal editorial.
Thank you very much for suggesting again this point. We included as you suggested us, the certificate of approval from the Institutional Review Board to the journal editorial.

Reviewer 4 Report
The authors addressed all the required items. The value of the manuscript increased consequently.
I recommend publication.
Author Response
The authors addressed all the required items. The value of the manuscript increased consequently.
I recommend publication.
Thank you very much for these comments and for the recommendation.